# Types and Fecundity of Neotenic Reproductives Produced in 5-Year-Old Orphaned Colonies of the Drywood Termite, *Cryptotermes domesticus* (Blattodea: Kalotermitidae)

Wenjing Wu [†], Zhenyou Huang [†], Shijun Zhang, Zhiqiang Li *, Bingrong Liu, Wenhui Zeng and Chuanguo Xia

Guangdong Key Laboratory of Animal Conservation and Resource Utilization, Guangdong Public Laboratory of Wild Animal Conservation and Utilization, Institute of Zoology, Guangdong Academy of Sciences, Guangzhou 510260, China; wuwj@giz.gd.cn (W.W.); huangzyoo2@163.com (Z.H.); zhangsj@giz.gd.cn (S.Z.); liubr@giz.gd.cn (B.L.); zengwh@giz.gd.cn (W.Z.); xiachuanguo@giz.gd.cn (C.X.)

* Correspondence: lizq@giz.gd.cn
† These authors contributed equally to this work.

**Abstract:** Orphaned colonies of *Cryptotermes domesticus* readily produce replacement reproductives and continue propagation. In this study, we aimed to investigate the production and fecundity of neotenic reproductives in 5-year-old colonies of *C. domesticus* after orphaning. All 15 experimental colonies were successfully re-established by the neotenic reproductive pair. Three types of neotenic reproductives with various wing-bud lengths were observed: type I with micro wing buds, type II with short wing buds, and type III with long wing buds. Four patterns of pairs made up of these neotenics, namely, type I + type II, type I + type III, type II + type II, and type II + type III, exhibited reproductive capacities similar to those of the primary reproductive pair. We speculated that these neotenic reproductives were derived from various nymphal instars. The 5-year-old colonies had three instars of nymphs, with the majority being in the second instar, followed by the first. Thus, the combination of neotenic reproductives with short wing buds and micro wing buds was the dominant differentiation pathway of the orphaned colonies. After the removal of the original primary reproductive pair, the nymphs matured into neotenic reproductives and took over reproduction in the colony in 107.40 ± 15.18 days. This study highlights the importance of quarantine and routine inspection of wood, as well as the significance of early prevention and control of *C. domesticus* infestation in wood. Moreover, this study confirms the high differentiation and reproductive capacities of *C. domesticus*.

**Keywords:** *Cryptotermes domesticus*; orphaned colony; neotenic; reproductives; wing bud; fecundity

## 1. Introduction

Termite societies have an adaptable caste system. In the natal orphaned colony, alates (adultoids) or neotenics (immature individuals with juvenile characteristics) may develop into supplementary or replacement reproductives when the primary reproductives weaken or die [1]. The developmental origins of neotenic reproductives differ in different termite species; they can originate from larvae, nymphs (nymphoids), or workers (ergatoids) [2–4]. For example, when orphaned, the neotenics can be produced by *Reticulitermes speratus* (Kolbe) [5,6] and *Macrotermes gilvus* (Hagen) [7] colonies consisting of workers with or without nymphs, *Macrotermes carbonarius* (Hagen) [7] colonies consisting of nymphs or alates, and *Coptotermes gestroi* (Wasmann 1896) [8] colonies consisting of larvae. Moreover, the types and fecundity of supplementary or replacement reproductives are highly variable across species. Neotenics of *Reticulitermes chinensis* Snyder can be divided into five types according to the presence and length of wings and wing buds: wingscale, long wing bud, short wing bud, micro wing bud, and wingbudless forms [9,10]. Adultoids in *Macrotermes* are divided into three types: pseudoimagos with poor pigmentation and irregularly

broken wings, microimagos (dwarf alates) with shortened wings, or normal adultoids [3]. The occurrence of neotenics originating from nymphs of different instars exhibited differences in size in neotropical termites [11]. Through the preneotenic stage, ergatoid females differentiating from workers of the neotropical termite *Nasutitermes aquilinus* acquired reproductive capacity with the presence of terminal oocytes and other reproductive features [12]. Most male neotenics that emerged in orphaned colonies of *R. speratus* were inconspicuously mature; however, they exhibited developed gonads and participated in sexual reproduction [13]. Meanwhile, neotenics from 3-year-old orphaned colonies of *C. gestroi* with different wing-bud lengths and no eyes were non-functional and only developed primary oocytes in their ovaries or empty spermatheca [8]. These studies indicate that, in addition to the type of termite species, the age and composition of orphaned colonies may also influence the developmental pathway, type, and fecundity of supplementary or replacement reproductives.

The domestic drywood termite, *Cryptotermes domesticus* (Haviland) (Blattodea: Kalotermitidae), nests and feeds inside wood, for example, on structural timber, flooring, doors, furniture, dead trees, roots, logs, and cultivated trees and shrubs in houses or in the wild [14–16]. It is native to Southeast Asia but has been widely disseminated through the movement of infested goods, furniture, and plants to China (Guangdong, Guangxi, Hainan, Taiwan, and Yunnan Provinces), Japan, Australia, and islands in the Pacific Ocean, causing serious damage to wood [17–21]. *C. domesticus* propagates and reproduces via two methods: First, the primary reproductives shed their wings, mate, and oviposit to establish a colony after a dispersal flight. Second, if the colony loses its primary reproductives and becomes orphaned, replacement reproductives develop to either inherit the colony or split the parental colony into several colonies [22]. No dispersal flight is the most common pathway for new colony establishment in *C. domesticus*, posing numerous challenges to the control of infestations. Individuals in the 4-year-old colonies of *C. domesticus* can readily differentiate into functional replacement reproductives [23]. Moreover, the orphaned colony does not need to be large; it might be as small as five nymphs [24]. Thus, if the *C. domesticus* colony is not completely eradicated, replacement reproductives will easily form and spread, leading to new infestations. However, the knowledge of the differentiation pathways, types, and fecundity of neotenic reproductives in *C. domesticus* is limited.

To address these problems, we conducted experiments on 5-year-old colonies of *C. domesticus* initiated by primary reproductives in the laboratory. Furthermore, the formation of neotenic reproductives and their morphological characteristics were studied, and they were divided into various types. Moreover, the preoviposition and egg periods, as well as the number of offspring, were recorded. This study provides a reference for further research on the strategies to control *C. domesticus* infestations.

## 2. Materials and Methods

### 2.1. Termite Collection

Wooden blocks inhabited by *C. domesticus* were collected before dispersal flights in Zhanjiang, Guangdong Province, China (21°12′ N, 110°28′ E). The blocks were transferred to the laboratory in Guangzhou, Guangdong Province, China (23°05′ N, 110°10′ E).

### 2.2. Colony Building by the Primary Pairs

The wooden blocks infested with *C. domesticus* were placed in a glass tank (90 cm length, 50 cm width, and 70 cm height) covered with a ventilated lid in the laboratory, and the termites were reared at room temperature until the occurrence of alates and dispersal flights. A bottle of water was put inside the tank to maintain humidity. During the dispersal phase of the alates, healthy (infection-free) blocks of *Parkia* sp. (7 cm length, 5 cm width, and 4.5 cm height) with a hole (0.25 cm diameter and 1.5 cm depth) were placed in the glass tank for the alates to feed and nest inside after flying, shedding their wings, and mating.

Successfully mated pairs became the primary reproductives and bred new colony members inside the *Parkia* sp. blocks. The individuals in these blocks were separately

reared in a small glass container (12 cm diameter and 12 cm height) at room temperature for 5 years.

*2.3. Orphaning Experiment*

After 5 years, 15 surviving colonies contained a queen, king, eggs, 7–16 larvae and/or pseudergates, 18–65 nymphs, and 1–4 soldiers. Next, both primary reproductives were removed from the *Parkia* sp. blocks. Then, 20 similar-sized individuals were selected randomly from the remaining members of the orphaned colony and transferred to new chambers. The chamber consisted of two blocks (7.5 cm length, 4.5 cm width, and 1.5 cm height), one of which had a hole (0.25 cm diameter and 1.5 cm depth). The termites were placed in the hole with a small piece of *Parkia* sp., and the hole was sealed with a cover slip for easy observation. All individuals in the wooden blocks were reared at room temperature in the same glass tank described in Section 2.2, with a ventilated lid and a bottle of water to maintain humidity.

*2.4. Observation of the Development of Orphaned Colonies*

The orphaned colonies were monitored daily for 140 days through the cover slip to record the incidence and quantity of neotenic reproductives, as well as the times of oviposition and larval hatching. After the formation of neotenic reproductives, they were observed once a month for 12 months to record the changes in the wing buds of the neotenic reproductives. At the end of the 12-month observation period, the chamber was dissected to assess the quantity of oviposition and larvae. The neotenic reproductives were observed under an Olympus SZ61 stereo microscope (Tokyo, Japan) after 1 year, and their wing-bud length and number of antennal articles were measured. The original statistics of the 15 experimental colonies are presented in Table A1. The data of emergence time, wing-bud length, and antennal articles of neotenic reproductives were used as input variables for cluster analysis (hierarchical and K-means cluster analysis) in IBM SPSS Statistics 26 (SPSS, Chicago, IL, USA). The types of neotenic reproductives were divided based on the output number of clusters. The total numbers of eggs oviposited and live larvae per neotenic reproductive type, measured after 1 year, were analyzed using the independent-samples nonparametric Kruskal–Wallis test in SPSS. All results were expressed as the mean ± SD, and $p < 0.05$ was considered significant.

## 3. Results

*3.1. Emergence of Neotenic Reproductives*

Neotenic reproductives were obtained in all ($n = 15$) 5-year-old orphaned colonies. The first neotenic was observed from 7 to 12 days after orphaning (average $9.00 \pm 1.41$ days), with 26.66% appearing on days 8 and 9 (Figure 1). The second neotenic emerged between days 13 and 21 (average $15.93 \pm 2.43$ days), with 26.67% appearing on day 14 (Figure 1). The interval between the first and second neotenics was 5–11 days.

*3.2. Morphological Observation*

The structure of the abdominal sternite in the neotenic reproductives was similar to that in primary reproductives (Figure 2). In *C. domesticus*, sex determination of neotenic reproductives could be performed based on the shape of their seventh abdominal sternite. The female neotenics had a posteriorly enlarged seventh abdominal sternite, covering the eighth and ninth sternites. However, in male neotenics, the seventh abdominal sternite was barely enlarged, and a pair of styli appeared in the center of the ninth sternite without segments. The wing buds of neotenic reproductives were less than half the length of those of primary reproductives and were difficult to remove. The neotenic reproductives had underdeveloped compound eyes and 12–15 antennal articles (Table 1). The head shell color of the neotenic reproductives was darker than that of nymphs, and it gradually deepened with growth. The color of the head, thorax, abdomen, feet, antennae, and wing buds

gradually deepened and became pale yellow; however, it remained lighter than that in primary reproductives.

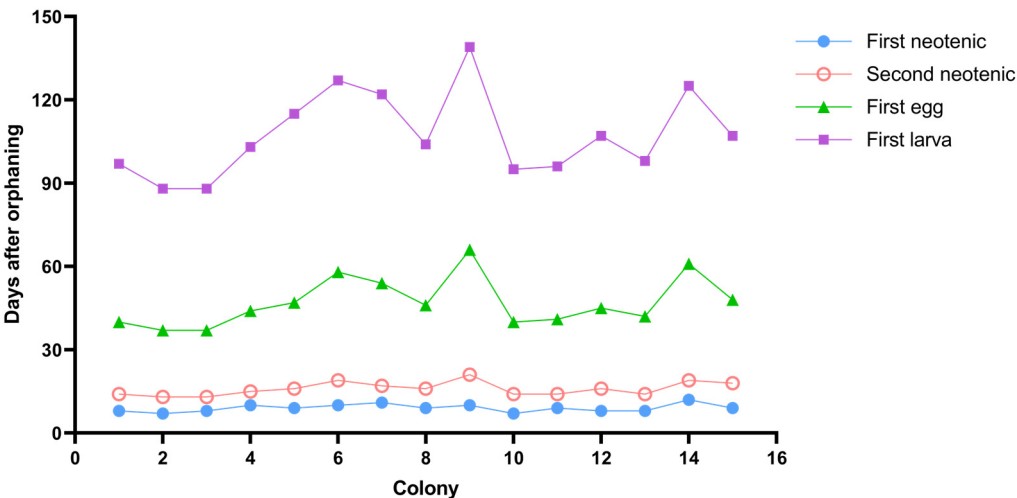

**Figure 1.** The emergence time of the first neotenic, second neotenic, first egg, and first larva in each experimental colony of *Cryptotermes domesticus*.

**Table 1.** Features of neotenic reproductives of *Cryptotermes domesticus*.

| Types | Number of Neotenic Reproductives | | | Wing-Bud Length (mm) | | Number of Antennal Articles | |
|---|---|---|---|---|---|---|---|
| | Total | First Neotenic | Second Neotenic | Range | Mean ± SD * | Range | Mean ± SD * |
| I Micro wing bud | 10 | 0 | 10 | 0.07–0.15 | 0.11 ± 0.02 c | 12–13 | 12.50 ± 0.82 c |
| II Short wing bud | 18 | 13 | 5 | 0.23–0.50 | 0.34 ± 0.07 b | 12–14 | 13.50 ± 0.62 b |
| III Long wing bud | 2 | 2 | 0 | 0.74–0.83 | 0.79 ± 0.06 a | 14–15 | 14.50 ± 0.71 a |

* Different letters indicate significant differences ($p < 0.05$).

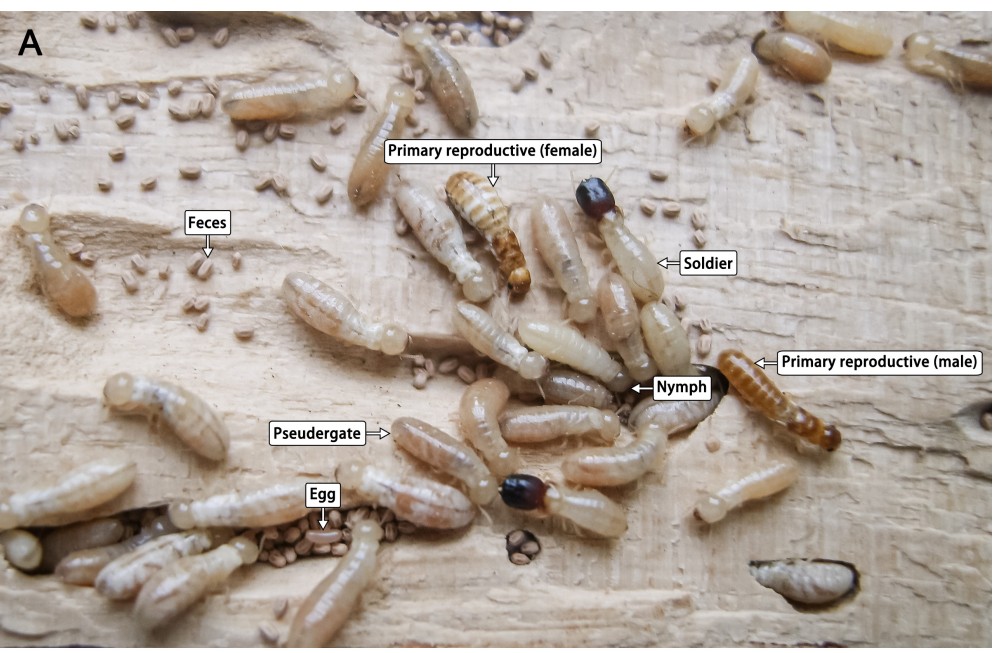

**Figure 2.** *Cont.*

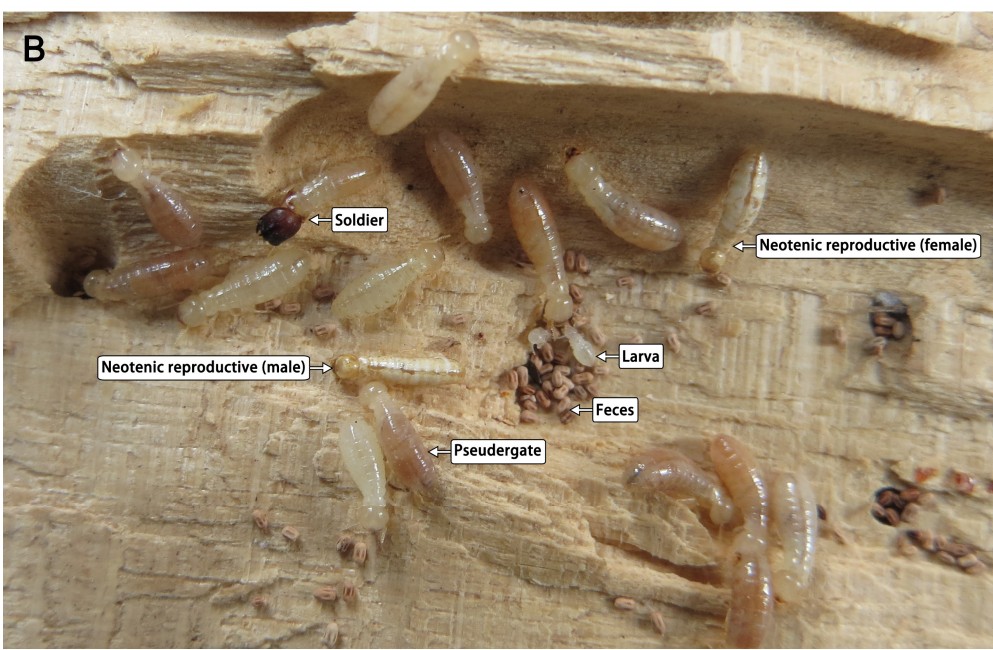

**Figure 2.** Colonies established by (**A**) primary reproductives and (**B**) neotenic reproductives of *Cryptotermes domesticus*.

### 3.3. Types and Composition of Neotenic Reproductives

All neotenic reproductives produced in the orphaning experiments possessed a pair of wing buds, but with different lengths. The clustering results indicated that the neotenic reproductives could be classified into three types based on their wing-bud length (Figure 3; Table 1): type I, with micro wing buds, which were just tiny protrusions posteriorly extending from the mesonotum and metanotum; type II, with short wing buds that extended into the first abdominal segment and were seen above the mesonotum and metanotum; and type III, with long wing buds that protruded beyond the first abdominal segment but exhibited less than half that the length of the wing buds of primary reproductives. All three forms had the wing buds stacked along either side of the meso- and metanota, and they only differed in terms of their length (Figure 3). The wings of type I and II neotenic reproductives were a similar color to their bodies. The wings of type III neotenic reproductives were darker in color than those of types I and II.

Type I with micro wing buds was not observed in any of the 15 first-occurring neotenic reproductives; however, 13 of them were type II with short wing buds (86.67%), and 2 were type III with long wing buds (13.33%). The first neotenic reproductives were predominantly type II with short wing buds. However, none of the 15 second-occurring neotenic reproductives were of type III with long wing buds; 10 and 5 were of type I with micro wing buds (66.67%) and type II with short wing buds (33.33%), respectively. The second neotenic reproductives were predominantly of type I with micro wing buds.

The combination patterns of the first and second neotenic reproductives in orphaned colonies were studied. Four different combination patterns were observed. The most common combination was type I + type II (nine pairs; 60.00%), followed by type II + type II (four pairs; 26.67%), type I + type III (one pair; 6.67%), and type II + type III (one pair; 6.67%) (Table 2).

### 3.4. Fecundity of Neotenic Reproductives

In the orphaned colonies, all neotenic reproductives could reach maturity, and females could reproduce posterity. Eggs were laid 24–45 days (average 31.13 ± 6.55 days) after the formation of male and female neotenic reproductives (Figure 1). The eggs required 51–73 days (average 60.33 ± 6.76 days) to develop into larvae (Figure 1). In the orphaned colonies, the initial times of the first neotenic reproductives, second neotenic reproductives,

egg laying, and larva hatching did not overlap. Extending the first three of these activities would prolong the hatching time.

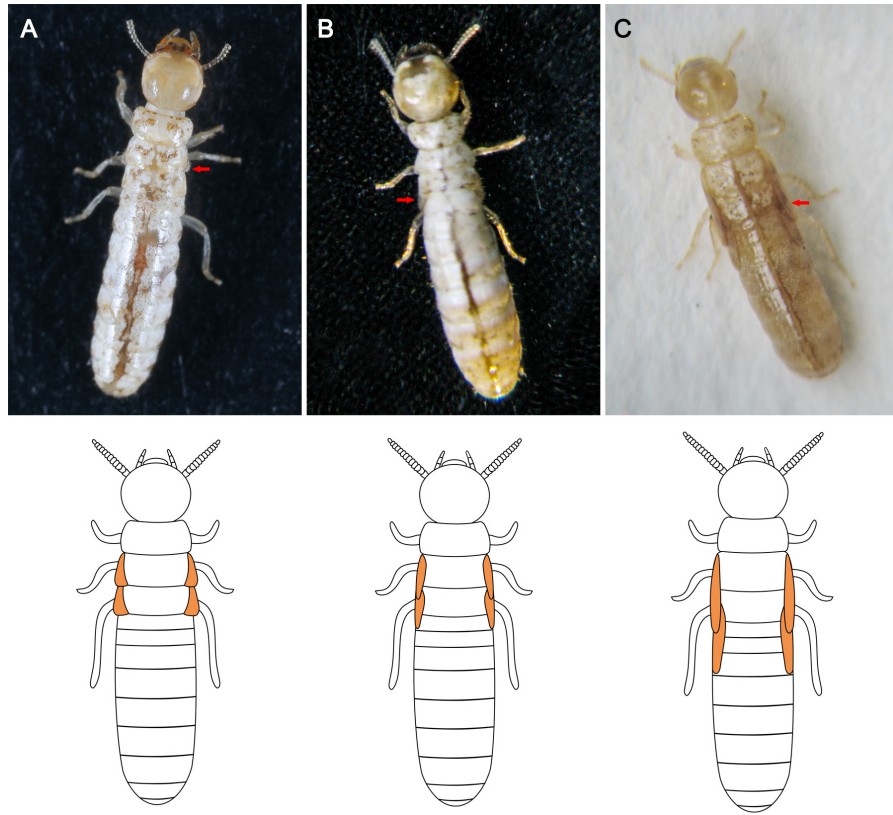

**Figure 3.** Types of neotenic reproductives of *Cryptotermes domesticus* with different wing-bud lengths: (**A**) Type I with micro wing buds. (**B**) Type II with short wing buds. (**C**) Type III with long wing buds. The red arrows indicate the wing buds. The schematic diagrams of the wing buds are provided below the photographs.

**Table 2.** Fecundity of neotenic reproductives of *Cryptotermes domesticus*.

| Combination Pattern | Total Number | Preoviposition Period (day) (Mean ± SD) | Egg Period (Day) (Mean ± SD) | After One Year | | | |
| | | | | Egg Number | | Number of Larvae | |
| | | | | Total | Mean ± SD | Total | Mean ± SD |
|---|---|---|---|---|---|---|---|
| I + II | 9 | 32.67 ± 6.26 | 63.44 ± 6.15 | 13 | 1.44 ± 0.88 | 24 | 2.67 ± 0.71 |
| I + III | 1 | 26 | 55 | 1 | 1 ± 0 * | 4 | 4 ± 0 |
| II + II | 4 | 30.75 ± 7.89 | 57.00 ± 5.48 | 3 | 0.75 ± 0.96 | 13 | 3.25 ± 0.96 |
| II + III | 1 | 24 | 51 | 1 | 1 ± 0 | 5 | 5 ± 0 |

* Since these patterns had only one repeat, SD is written as 0.

One year after the formation of neotenic reproductives, the wood chambers were dissected. The numbers of eggs and larvae were measured, which averaged 1.20 ± 0.86 and 3.07 ± 0.96, respectively (Figure 4). No soldiers appeared until then. In three experimental colonies, all of the eggs developed into larvae. Most colonies included one egg (46.67%) and two or three larvae (66.67%). Table 2 shows the fecundity of different combination patterns. No significant differences were observed in the preoviposition period (H = 4.143, $p = 0.246$), egg period (H = 6.405, $p = 0.093$), egg numbers (H = 1.846, $p = 0.605$), or numbers of live larvae (H = 5.487, $p = 0.139$) among the four combination patterns.

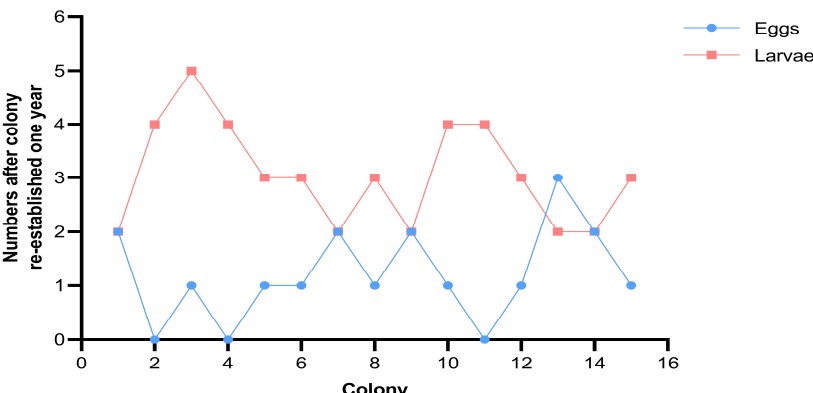

**Figure 4.** The numbers of eggs and larvae one year after colony re-establishment of *Cryptotermes domesticus*.

## 4. Discussion

This study found that *C. domesticus* colonies had three different types of neotenic reproductives, including type I with micro wing buds, type II with short wing buds, and type III with long wing buds. The 5-year-old orphaned colonies could produce four combination patterns of neotenic reproductives: type I + type II, type I + type III, type II + type II, and type II + type III. Neotenic reproductives from all combination patterns exhibited good fecundity and fertility, which were similar to those of primary reproductives.

In total, five types of neotenic reproductives were observed in *R. chinensis*: wingscale, long wing buds, short wing buds, micro wing buds, and wingbudless forms [9,10]. In this study, only three types of neotenic reproductives were observed in *C. domesticus*, lacking wingscale and wingbudless forms. In *R. chinensis*, alates that lost their wings and reproduced inside the natal nest (i.e., did not conduct a nuptial dispersal flight) developed into wingscale adultoids [9,10]. However, *C. domesticus* colonies in Guangzhou were matured after 7 years and began to produce alates at room temperature [25]. There was no foundation for the development of wingscale forms in the 5-year-old orphaned colonies, because they were too immature to produce alates in *C. domesticus*. Wingscale forms could be obtained using 7-year-old orphaned colonies. Wingbudless forms were derived from the larvae in *R. chinensis* [9,10]. Although larvae were present in 5-year-old *C. domesticus* colonies, they had to compete with the nymphs for the opportunity to grow into neotenic reproductives. Even if they grew into neotenics, neotenics from nymphs were more powerful and aggressive. Therefore, nymphs or neotenics developed from nymphs had a higher chance of surviving confrontation. In a previous study, 2.25-year-old orphaned colonies could produce replacement reproductives [23]. Wingbudless forms could be produced from 2.25- to 4-year-old orphaned colonies. However, this should be further investigated in future studies.

Wing buds were observed on three different types of neotenic reproductives. In these orphaned colonies, only the nymphs had wing buds. This indicated that the nymphs may be the origin of primary differentiation for neotenic reproductives in orphaned colonies of *C. domesticus* that are 5 years old. Similarly, three instars of nymphs (third, fourth, and fifth) could become neotenics in the neotropical termite *Silvestritermes euamignathus*, also differentiated by their wing-bud lengths [26].

Studies have revealed that the genus *Cryptotermes* exhibits various numbers of nymphal instars. Five instars are present in the nymph developmental pathway of *C. secundus* [27,28], whereas three instars are present in *C. cavifrons* [29] and *C. dudleyi* [30]. In this study, three types of neotenic reproductives with wing buds were observed in *C. domesticus*. This indicates that its nymphs may have three instars: the first-instar nymph with micro wing buds, the second-instar nymph with short wing buds, and the third-instar nymph with long wing buds. Furthermore, during the 1-year observation period, no growth or regression in the wing buds of neotenic reproductives was observed. This indicates that these wing bud

types cannot be converted into one another. In addition, the neotenic reproductives could not shed their wing buds and could not mature into wingscale forms. This further suggests that the origins of the wingscale forms were different from those of types I, II, and III.

In addition to the length of the wing buds, antennal articles can be considered as another feature to distinguish nymph instars. Type I, II, and III neotenic reproductives had different numbers of antennal articles (12–13, 13–14, and 14–15, respectively) and fewer than primary reproductives (16 articles) [18]. In *C. dudleyi*, the first, second, and third nymphal instars contained 10–11, 11–12, and 12–13 articles, respectively. This implies that the number of antennal articles increases in tandem with instar development [30]. The number of antennal articles in nymphs and in neotenic reproductives can be correlated, and this number can be used to classify nymph instars in *C. domesticus*.

Among the 30 neotenic reproductives, type II with short wing buds was the most common form (60.00%), followed by type I with micro wing buds (33.33%); type III with long wing buds was the least common form (6.67%). This indicates that neotenic reproductives with short wing buds may be the main differentiation pathway for 5-year-old orphaned colonies of *C. domesticus*. In addition, it indicates that the nymphal development in the 5-year-old *C. domesticus* colonies was inconsistent, probably due to the microenvironment, which has been reported to influence reproductive development [27]. The majority of nymphs in the 5-year-old colonies molted to the second instar, although some remained in the first instar and a few in the third instar. This indicates that the quantity, instar, and gender of nymphs affect the formation of neotenic reproductives and further affect their combination patterns. Additionally, nymph differentiation occurred in sequence, with older nymphs the first to convert into neotenic reproductives. The type III + type III pattern is challenging to form because of the small number of third-instar nymphs. Moreover, type II neotenic reproductives with short wing buds will emerge first, since the second-instar nymphs are the most common in the 5-year-old colonies, making it challenging to develop the type I + I pattern.

The fecundity of neotenic reproductives was roughly similar to that of primary reproductives because of the approximate similarity of their egg period and quantity of larvae after 1 year. The preoviposition period for neotenic reproductives was 24–45 days after the emergence of the second neotenic reproductive, and the larvae were hatched from the eggs after 51–73 days. Compared with primary reproductives with preoviposition and egg periods of 7–16 and 46–71 days, respectively [22], the oviposition of neotenic reproductives was delayed. However, the egg periods were similar. Neotenic reproductives derived from nymphs in orphaned colonies were used to compensate for the absence of primary reproductives. Male and female neotenic reproductives appeared sequentially because they were immature and needed time to develop before reproducing. On the other hand, the primary reproductives were completely developed and mated shortly after a nuptial dispersal flight. Therefore, neotenic reproductives had a longer preoviposition period than primary reproductives. Because all eggs deposited by them were fertilized, the egg-hatching period was similar for primary and neotenic reproductives. In 1-year-old colonies, neotenic reproductives produced 2–5 larvae, which was similar to the numbers produced by primary reproductives (3–8 larvae) [22].

## 5. Conclusions

This study found that 5-year-old colonies of *C. domesticus* had three nymph instars, which could be recognized by features such as wing-bud length and number of antennal articles. All three nymphal instars could differentiate into neotenic reproductives, resulting in three types of neotenic reproductives with fecundity similar to that of primary reproductives. Older nymphs had priority for differentiation and exhibited a higher rate of survival. In the 5-year-old colonies of *C. domesticus*, most nymphs reached the second instar with short wing buds, allowing for the establishment of a new breeding colony in 88–139 days after orphaning. Older colonies may divide more quickly and cause greater damage. Although challenging, fumigating wooden structures in buildings is the most

effective approach to prevent drywood termite infestations. The remaining individuals may inherit the original colony and will probably differentiate into new colonies and continue to cause harm. Therefore, when using pesticides for local treatment, it is advisable to destroy all members of the termite colony at once. The wooden components of buildings, as well as the dry wood of old trees, must be inspected on a regular basis for termite-induced damage and other signs of degradation. Essential measures to prevent and control *C. domesticus* infestation include strengthening the wood quarantine, treating damage as soon as possible, and avoiding the development of replacement reproductives.

**Author Contributions:** Conceptualization, W.W. and Z.H.; data curation, C.X.; funding acquisition, Z.H. and Z.L.; investigation, B.L.; methodology, S.Z.; resources, W.Z.; visualization, W.W.; writing—original draft, W.W. and Z.H.; writing—review and editing, W.W. and Z.L. All authors have read and agreed to the published version of the manuscript.

**Funding:** This research was funded by the National Natural Science Foundation of China (grant number 31172163), the Natural Science Foundation of Guangdong Province (grant number 010085), and the GDAS Special Project of Science and Technology Development (grant number 2017GDASCX-0107).

**Institutional Review Board Statement:** Not applicable.

**Data Availability Statement:** The data presented in this study are available in this article.

**Acknowledgments:** The authors are grateful to Xing Qian for gathering experimental materials and offering technical assistance. The authors acknowledge Wenming Lan for his help in acquiring the *C. domesticus*-infected wooden blocks. The authors would like to thank Zhengming Ping for identifying *C. domesticus*. The authors would like to thank Jian Hu for providing valuable criticism on this work. The authors also thank Zhenhua Liu for helping in photography.

**Conflicts of Interest:** The authors declare no conflicts of interest.

## Appendix A

**Table A1.** The original statistics of 15 experimental colonies of *Cryptotermes domesticus*.

| Colony | First Neotenic | | | Second Neotenic | | | Preoviposition Period (Day) | Egg Period (Day) | After One Year | |
|---|---|---|---|---|---|---|---|---|---|---|
| | Emergence Time (Day) | Wing-Bud Length (mm) | Number of Antennal Articles | Emergence Time (Day) | Wing-Bud Length (mm) | Number of Antennal Articles | | | Egg Number | Number of Larvae |
| 1 | 8 | 0.42 | 14 | 14 | 0.13 | 13 | 26 | 57 | 2 | 2 |
| 2 | 7 | 0.35 | 14 | 13 | 0.23 | 13 | 24 | 51 | 0 | 4 |
| 3 | 8 | 0.74 | 15 | 13 | 0.3 | 14 | 24 | 51 | 1 | 5 |
| 4 | 10 | 0.33 | 13 | 15 | 0.14 | 13 | 29 | 59 | 0 | 4 |
| 5 | 9 | 0.38 | 14 | 16 | 0.11 | 12 | 31 | 68 | 1 | 3 |
| 6 | 10 | 0.4 | 14 | 19 | 0.09 | 13 | 39 | 69 | 1 | 3 |
| 7 | 11 | 0.31 | 13 | 17 | 0.07 | 12 | 37 | 68 | 2 | 2 |
| 8 | 9 | 0.36 | 14 | 16 | 0.25 | 13 | 30 | 58 | 1 | 3 |
| 9 | 10 | 0.37 | 13 | 21 | 0.09 | 12 | 45 | 73 | 2 | 2 |
| 10 | 7 | 0.83 | 14 | 14 | 0.15 | 13 | 26 | 55 | 1 | 4 |
| 11 | 9 | 0.38 | 14 | 14 | 0.26 | 14 | 27 | 55 | 0 | 4 |
| 12 | 8 | 0.33 | 13 | 16 | 0.1 | 12 | 29 | 62 | 1 | 3 |
| 13 | 8 | 0.5 | 14 | 14 | 0.11 | 12 | 28 | 56 | 3 | 2 |
| 14 | 12 | 0.3 | 13 | 19 | 0.25 | 12 | 42 | 64 | 2 | 2 |
| 15 | 9 | 0.35 | 14 | 18 | 0.1 | 13 | 30 | 59 | 1 | 3 |

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
