# Peer review of "Types and Fecundity of Neotenic Reproductives Produced in 5-Year-Old Orphaned Colonies of the Drywood Termite, Cryptotermes domesticus (Blattodea: Kalotermitidae)"

_diversity, doi:10.3390/d16040250_

Round 1
Reviewer 1 Report
Comments and Suggestions for Authors
The manuscript by Wu et al. shows an interesting morpho-functional approach on neotenic differentiation after orphaning in the drywood termite C. domesticus. The authors experimentally orphaned 15 colonies and observed the origin of replacement reproductives, which varied among colonies considering the nymphal instar of origin. They also followed the first oviposition and larvae hatch of the colonies then headed by neotenics. Though I think it is an interesting study, it may some questions to solve prior to a possible acceptance for publication. In addition to that questions, though I am not a native English-speaker, I think that the text needs English editing by a native-speaker.
1. Were the first neotenic male or female? What about the second neotenic that appeared in the orphaned colony?
2. Were those two neotenics the unique ones that appeared in the orphaned colonies? I ask that because it is known for some termite species that many neotenic reproductives differentiate and fight to inherit the natal nest, especially in non-Termitidae species.
3. In addition to the prior question, how do the authors compare the fecundity and found it comparable between neotenics and primaries? I had this doubt and if it is because of the egg numbers found after one year, it should be clarified. If not, how so?
4. The 15 orphaned colonies appeared to be well-syncronized with few variance on the appearance of neotenic reproductives. Were they kept at the same conditions since its start with primaries until the orphaning and after that?
Below, some specific comments.
Line 12: rewrite the sentence “...produce supplementary reproductives to spread their damage.” because supplementary reproductives have a different meaning from the termite life history point of view.
Line 17-18: it would be interesting to have a brief explanation of each pattern considering four different capacities found in the study. The way it is written is not clear.
Lines 40-41: after the sentence “The types and fecundity of supplementary or replacement reproductives are also highly variable across species” it would be interesting to cite the following studies characterizing, respectively, the occurrence of neotenics originating from nymphs of different instars and from workers showing differences in fecundity, in two neotropical termite species.
Haifig et al. 2016. Unrelated secondary reproductives in the neotropical termite Silvestritermes euamignathus (Isoptera: Termitidae). Naturwissenschaften, v. 103, p. 2-8 http://dx.doi.org/10.1007/s00114-015-1325-0
Silva et al. 2019 Ergatoid reproductives in the Neotropical termite Nasutitermes aquilinus (Holmgren) (Blattaria: Isoptera: Termitidae): developmental origin, fecundity, and genetics. Insect Science https://doi.org/10.1111/1744-7917.12727
Lines 66. I did not understand the sentence.: “Because merely five nymphs from four-year-old colonies of C. domesticus can readily produce neotenic reproductives with reproductive function” Do the authors mean that colonies with that age have individuals ready to become functional secondary reproductives?
Line 112. Could the authors explain why do they use cluster analysis for analyzing neotenic data?
Line 114-115. Were the total number of eggs and larvae calculated considering which period interval? It would also help to understand table 2 and line 188. Considering that, was it after one year?
Line 188-189. Start the sentence as: No significant differences were observed in pre-oviposition period… The way it is written that Kruskal-Walis test found is awkward.
Line 197. …produce four combination patterns of neotenic occurrence, such as…
Line 220-223. Consider to cite HAIFIG, I.; COSTA-LEONARDO, A. M. Caste differentiation pathways in the Neotropical termite Silvestritermes euamignathus (Isoptera: Termitidae). Entomological Science (Tokyo), v. 19, p. 174-179, 2016. http://dx.doi.org/10.1111/ens.12201,which shows that 3 instars of nymphs (3rd, 4th and 5th) were able to become neotenics in a neotropical termite species, differentiated by their wing bud lengths.
Comments on the Quality of English Language
Though I am not a native English-speaker, I think that the text needs English editing by a native-speaker.
Reviewer 2 Report
Comments and Suggestions for Authors
This study demonstrates colony neotenic development (three wing bud categories) after in five-year-old Cryptotermes domesticus colonies that were observed for an additional year in observation blocks. After the observation blocks were “dissected” the number of eggs and larvae produced by the neotenics was reported.
Why were not all the colony members (pseudergates, nymphs, and soldiers) counted at five years and after the sixth year when the blocks were dissected? These data would provide the overall demographics of these C. domesticus colonies.
Three important references related to C. domesticus colony development could not be located by me. Please proved Digital Object Identifiers:
Qian Xing, Huang Zhenyou, Zhong Junhong, Dai Zirong, Liu Bingrong, Xia Chuanguo, Huang Haitao, Xia Feng, Yang Ruihai, and Zhang Ruilin. "The formation and development of a new colony of Termite truncatula." Natural Enemies of Insects, no. 03 (2005): 118-26.
Huang, Zhenyou, Xing Qian, Junhong Zhong, Jian Hu, Zhiqiang Li, Qiujian Li, and Bingrong Liu. "Influence of Colony Age on Production and Fecundity of Replacement Reproductives in the Dry Wood Termite Cryptotermes Domesticus (Isoptera: Kalotermitidae)." Sociobiology 58, no. 1 (2011): 77-84.
Huang Zhenyou, Qian Xing, Zhong Junhong, Dai Zirong, Liu Bingrong, Xia Chuanguo, Huang Haitao, Xia Feng, Yang Ruihai, and Zhang Ruilin. "The formation cycle of primitive breeding ants of Termite truncatula." Insect Knowledge, no. 05 (2005): 528-31.

Round 2
Reviewer 2 Report
Comments and Suggestions for Authors
No further comments.